# Perfecting the Life Clock: The Journey from PTO to TTFL

**DOI:** 10.3390/ijms24032402

**Published:** 2023-01-26

**Authors:** Weitian Li, Zixu Wang, Jing Cao, Yulan Dong, Yaoxing Chen

**Affiliations:** 1College of Veterinary Medicine, China Agricultural University, Haidian, Beijing 100193, China; 2Department of Nutrition and Health, China Agricultural University, Haidian, Beijing 100193, China

**Keywords:** circadian rhythm, evolution, biological clock, TTFL, PTO

## Abstract

The ubiquity of biological rhythms in life implies that it results from selection in the evolutionary process. The origin of the biological clock has two possible hypotheses: the selective pressure hypothesis of the oxidative stress cycle and the light evasion hypothesis. Moreover, the biological clock gives life higher adaptability. Two biological clock mechanisms have been discovered: the negative feedback loop of transcription–translation (TTFL) and the post-translational oscillation mechanism (PTO). The TTFL mechanism is the most classic and relatively conservative circadian clock oscillation mechanism, commonly found in eukaryotes. We have introduced the TTFL mechanism of the classical model organisms. However, the biological clock of prokaryotes is based on the PTO mechanism. The Peroxiredoxin (PRX or PRDX) protein-based PTO mechanism circadian clock widely existing in eukaryotic and prokaryotic life is considered a more conservative oscillation mechanism. The coexistence of the PTO and TTFL mechanisms in eukaryotes prompted us to explain the relationship between the two. Finally, we speculated that there might be a driving force for the evolution of the biological clock. The biological clock may have an evolutionary trend from the PTO mechanism to the TTFL mechanism, resulting from the evolution of organisms adapting to the environment.

## 1. Introduction

Four hundred million years ago, the growth lines on the nautilus shell and the growth rings of corals were evidence of circadian rhythms visible to the naked eye [1,2]. Cyanobacteria (*Synechococcus elongatus*) already had circadian rhythms three billion years ago and thus gained adaptive advantages [3]. The circadian rhythm is thought to give organisms greater adaptability in a rhythmic environment. It adjusts the metabolism and physiological activities of the body by making periodic predictions of environmental factors such as light, temperature, and humidity [4]. The predictable circadian rhythm exists across species from prokaryotes to eukaryotes [5]. In different species, the molecular composition of the biological clock is not conservative. The core of the eukaryotic biological clock is an autonomous oscillation system with negative feedback regulation. This oscillation system includes positive and negative regulatory components [6,7]. In prokaryotes is a post-translational oscillation mechanism that does not involve transcription and translation. However, although significant progress has been made in understanding the physiological and molecular mechanisms of the biological clock, research on the evolution of the biological clock has been very limited. We collate the conjectures about the origin of the biological clock at this stage and sort out the possibility of promoting the evolution of the biological clock by discussing and comparing the differences and connections of the core oscillator mechanisms of different species.

## 2. The Origin of the Circadian Rhythm

There are two models for the molecular clock mechanism of organisms: the transcription–translation feedback loop (TTFL) and the post-translational oscillation mechanism (PTO). Different species have different oscillator structures. Cyanobacteria are the oldest and most primitive known species with circadian rhythms. Oscillators in cyanobacteria are composed of three proteins: Kai A, Kai B, and Kai C. There is evidence that the origin of the KaiA gene can be traced back to the origin of most cyanobacteria, around 3000 ± 500 Mya [8]. In the fungus *Neurospora crassa*, TTFL is composed of the frequency (FRQ) protein and WHITE/COLLAR complex (WC), while in plants, TTFL includes TOC1 and CCA1 [9,10]. Some species have the same TTFL components, such as the Period protein (PER) in flies and humans. Although the role of this protein is not the same between the two species, it implies that more than 500 million years ago, insects and mammalian common ancestors had a similar circadian clock mechanism [11,12,13]. We still do not know the origin of the biological clock, but several recent studies have provided new insights.

### 2.1. Selective Stress Hypothesis of the Oxidative Stress Cycle

Peroxiredoxin (PRX or PRDX) is highly conserved among different species, and its oxidative state exhibits circadian rhythm oscillations in humans, mice, and seaweeds, reflecting reactive oxygen species’ (ROS) endogenous rhythm [14,15]. Because almost all organisms have PRX, some scholars have proposed that the metabolic rhythm of peroxidase is conservative in the three major systems of archaea, bacteria, and eukaryotes [16,17]. This makes one wonder whether the oxidative activity is the original factor that promoted the appearance of the biological clock.

It is generally believed that prokaryotes have the most advanced circadian mechanism, and they have further changed life on Earth through the Great Oxygenation Event (GOE) [18]. Since the existing *Synechococcus elongatus* species has a strong circadian rhythm in photosynthesis and almost all other metabolic activities, it is an important model organism for studying the circadian rhythm of prokaryotes. The cyanobacteria rhythm is based on the slow phosphorylation and dephosphorylation of KaiA, KaiB, and KaiC, which together form a self-sustaining delayed negative feedback loop [19,20]. *S. elongatus* shows a second post-translational rhythmic oscillation, which involves the superoxidation of the redox-regulatory PRX protein; furthermore, it has a rhythmic redox-regulating capacity in archaea (*Halobacterium salinarum*), which is now generally considered to be in response to the surge in oxidation in the GOE [16]. As there are approximately 24 h of oxidation–reduction cycles in all areas of life, the hypothesis that cell rhythms have a common molecular origin is increasingly credible. The main cellular role of redox is to eliminate metabolic byproducts of toxicity, such as ROS. It is believed that the ability to survive in the oxidative stress cycle after the emergence of aerobic life is a selective advantage [16].

### 2.2. Light Evasion Hypothesis

For cyanobacteria, being able to predict the arrival of light at dawn and being prepared for it has a survival advantage over being able to respond to light and dark changes. This prediction prepares for protective reactions to harmful components in light, such as ultraviolet (UV) light [21]. There is still a “light evasion” hypothesis. This hypothesis holds that the circadian rhythm evolved due to the daily periodic selection pressure of light and dark, under which light became a factor that impaired growth [22,23].

The ability to respond to light is an essential feature of the biological clock, which is synchronized with the core biological clock by sensing non-visually sensitive light-sensitive retinal ganglion cells (ipRGCs) located on the retina to project light onto the suprachiasmatic nucleus (SCN) of the hypothalamus [24]. This allows the organism to have the ability to predict the periodic light and dark changes caused by the rotation of the Earth, so that it can arrange its physiological activities and various functions according to the change of the photoperiod. For example, cyanobacteria can ensure incompatible nitrogen fixation and separation of photosynthesis through rhythmic oscillations corresponding to the photoperiod [25]. At the same time, predicting the photoperiod can enable plants to begin some important processes of photosynthesis in advance, improving the efficiency of photosynthesis [26]. The lack of photoreceptors CCA1 and LHY in *Arabidopsis* leads to a decline in their adaptability [27]. Processes such as DNA replication are sensitive to UV light, and transferring them to night has obvious advantages for the organism. The unicellular algae (*Chlamydomonas reinhardtii*) were found to be most sensitive to UV radiation at the end of the day and night, and the highest sensitivity was recorded during mitosis [22]. All kinds of evidence show that the perception of the photoperiod is an important part of species’ adaptation.

There are many harmful components in natural light, such as UV rays. In early life, if there is no corresponding countermeasure to resist light damage, it will inevitably be accompanied by a decline in competitiveness and adaptability. However, the energy of living organisms is limited. If the protection mechanism against natural light damage is always maintained, unnecessary energy will be consumed at night, which is not efficient for living organisms. Therefore, individuals who maintain their protective effect against natural light damage only during the day and not at night or at low levels of light are more competitive than those who maintain this protection throughout the day. It can save energy for other areas, such as feeding, digestion, or reproduction. This hypothesis suggests a possible origin of the biological clock system.

### 2.3. Adaptability of the Biological Clock

Functional circadian clocks provide adaptive advantages for organisms (Figure 1). First, they coordinate internal physiological activities producing internal advantages. Second, they synchronize internal physiological activities with external periodic environmental changes to form external advantages. The biological clock has two states: entrainment in the external periodic cycle and free-running in the non-periodic environment. Even organisms living in non-periodic environments, such as caves and deep seas, still have a periodic rhythm of approximately 24 h. This shows that intrinsic advantages are just as important. Therefore, the circadian rhythm has a far-reaching impact on the adaptability of living beings, which far exceeds the synchronization of the external environment [28].

## 3. Evolution of the Biological Clock Mechanism

Almost all living organisms have a functional circadian clock system. There are two mechanisms of the circadian clock system: the TTFL and the PTO mechanism.

### 3.1. Transcription–Translation Feedback Loop 

TTFL is the most classic oscillation mechanism. Jeffery Hall, Michael Rosbash, and Paul Hardin first proposed a model of a negative feedback loop for transcription and translation. This model was first described as the expression of the Period gene to increase the PER protein. When the PER protein reaches a certain concentration, it will feedback-inhibit the Period gene transcription, thereby reducing the PER protein. The decline in the PER protein level again initiates the transcription of the Period gene, forming a periodic reciprocation [29]. Subsequently, timeless (*Tim*), doubletime (*Dbt*), cryptochrome (*Cry*), clock circadian regulator (*Clock*), brain and muscle arnt-like 1 *(Bmal1*), cycle (*Cyc*), and other genes were found in *Drosophila* and mammals. The description of the TTFL model has been further confirmed and has become a recognized core clock oscillation mechanism. Although molecular clocks have different origins and are not conserved among different species, eukaryotes have adopted common design principles. This model relies on a TTFL, in which feedback of the protein products of clock genes periodically regulates their expression and drives rhythmic output pathways and physiology [6,30,31].

The circadian clocks of classic model organisms such as *N. crassa*, *Arabidopsis*, *Drosophila*, zebrafish (*Danio rerio*), birds, and mice (*Mus musculus*) are all TTFL mechanisms. Due to the evolutionary distance of these species, the TTFL mechanism is considered a well conserved biological clock oscillation mechanism [32]. The TTFL system consists of positive and negative regulators. Positive regulators are generally transcriptional activators containing PAS domains. They can bind to the promoter region encoding the negative regulator genes, activating their transcription. After translation, negative-regulator proteins inhibit the transcriptional activity of positive regulators in the nucleus, thereby suppressing the transcription of negative regulators themselves [33]. The positive regulators in *N. crassa* are white collar 1 (WC-1) and white collar 2 (WC-2), while the negative regulators are FRQ. The positive regulators in *Drosophila* are *Clock* and *Cyc*, while the negative regulators are *Per* and *Tim*. The positive regulators in birds are *Clock* and *Bmal1*, while the negative regulators are *Per2*, *Per3*, *Cry1*, and *Cry2*. The positive regulators in mice are *Clock* and *Bmal1*, while the negative regulators are *Per1*, *Per2*, *Per3*, *Cry1*, and *Cry2*. Zebrafish have positive and negative regulators similar to birds and mammals, but there are multiple copies of each regulator [5,7,34,35].

In the core oscillator of *N. crassa*, white collar complex (WCC), consisting of WC-1 and WC-2, drives transcription of the *Frq* gene as a transcription activator, and the translated FRQ protein forms a complex with FRQ-interacting RNA helicase (FRH) and casein kinase I (CKI). The time delay caused by these negative feedback factors during negative feedback inhibition of self-expression is the biological clock cycle [9,36,37]. In this process, the phosphorylation of the FRQ protein affects protein stability, which in turn affects the clock cycle [38,39,40,41]. The process of protein phosphorylation and dephosphorylation is a post-translational mechanism. This composition of the *N. crassa* biological clock also suggests a link between the TTFL and PTO mechanisms. However, the homologous genes of the FRQ protein of *N. crassa* in fungi are not universal, and the homologous genes of the WC-1 protein are more widely distributed in fungi. Therefore, the biological clock with FRQ-WCC as its core component is presumably not the only form of the biological clock in fungi. This reminds us that the biological clock can evolve independently, even in fungi.

The core oscillator of *Drosophila*’s biological clock is the most thoroughly studied. The heterodimer formed by CLOCK and CYC promotes the transcription of *Per* and *Tim* and other factors related to the circadian clock chain feedback loop. PER and TIM accumulate and form the heterodimer PER/TIM at night, entering the nucleus and promoting the phosphorylation of the CLOCK/CYC heterodimer, thereby inhibiting the vitality of the CLOCK/CYC dimer and reducing its affinity with DNA. During the day, CRY, a blue-light receptor (cryptocyanin), undergoes conformational changes after receiving light and spontaneously binds to TIM, causing ubiquitination of TIM, and thus, the decreased PER/TIM heterodimer facilitates the dimerization of CLOCK/CYC again. The inhibitory effect of the body no longer exists, so the vitality of the CLOCK/CYC dimer is restored, and a new cycle is started [42]. *Drosophila* circadian genes have high homology with mammalian circadian genes and are well-conserved [43].

The core oscillators of birds and mammals are the same, with only slight differences. BMAL and CLOCK form a heterodimer and bind to the E-box of the promoter region of the negative-regulator genes, promoting the transcription of the *Cry* and *Per* genes, and then translate into the clock proteins CRY and PER. CRY-PER binds together and enters the nucleus, affecting the CLOCK/BMAL1 heterodimer and inhibiting its own transcription [5]. Zebrafish have a similar TTFL loop [44]. Unlike mammals, teleost fish have an additional third genome duplication, so many zebrafish genes have different copies [45]. However, in evolution, many extra copies of genes may be lost. Some extra copies have similar functions, but others are different, and new functions are derived [46]. Three *Bmal* genes, three *Clock* genes, four *Per* genes, and six *Cry* genes have been found in zebrafish [46,47]. During the long-term evolution of species history, different copies of the circadian clock gene will produce different functions. Through high-throughput sequencing and the application of a large number of clock genome databases, the evolution of circadian clock genes is better described [48]. Multiple copies of these circadian clock genes will complicate the regulation mechanism of circadian clock rhythms and components, thus providing more opportunities for studying circadian clocks.

The core oscillator of plants consists of three closely related transcription–translation negative feedback loops: the core loop, the morning loop, and the evening loop. LHY, CCA1, and TOC1 form the central or core feedback loop. Both *Cca1* and *Lhy* are expressed in the morning. After their transcription and translation into proteins, they bind as the heterodimer CCA1/LHY inhibiting the expression of *Toc1*. At night, the expression levels of *Cca1* and *Lhy* decrease, and the expression of *Toc1* reaches a peak, inhibiting the expression of *Cca1* and *Lhy*, which causes them to oscillate [49,50]. *Cca1/Lhy* and *Prr9/Prr7* of the pseudo-response regulators (*Prr*) gene family form the morning cycle. PRR9 and PRR7 form a transcriptional repression complex to inhibit the expression of *Cca1* and *Lhy*, while CCA1/LHY inhibits the expression of *Prr9/7* [51,52]. Finally, the EC complex (consisting of LUX ARRHYTHMO, ELF3, and ELF4) and CCA1/LHY form the evening loop. The EC complex can promote the expression of *Cca1/Lhy*, while CCA1/LHY can inhibit the transcription of the EC complex [53,54]. The morning loop, the core loop, and the evening loop are interlinked and together form the core oscillator of the plant biological clock. Comparative transcriptome analysis showed that clock TTFL was conserved in many land plants. However, clock output gene expression is considered organ-, tissue-, or cell-specific to properly control the output and accommodate environmental fluctuations from diurnal or seasonal variations [55]. Compared with the core oscillator of an animal’s circadian clock, there are many details of the core oscillator of plants waiting to be discovered, but it still belongs to the TTFL mechanism.

### 3.2. Post-Translational Oscillation (PTO)

Although the above organisms have the same biological clock mechanism, the TTFL mechanism seems limited to eukaryotes, while the prokaryotic biological clock is not a TTFL mechanism. The core oscillator components of the circadian clocks of prokaryotes and eukaryotes do not have evolutionary homology because the cyanobacteria clock comprises KaiA, KaiB, and KaiC, and no similarity has been found among these three proteins’ source sequence in eukaryotes [56,57]. Therefore, the existing circadian clock system was generated by at least two independent evolutionary events [58].

First, it was discovered that algae still have a circadian photosynthesis cycle in the absence of nuclear transcription [59]. This implies that there is a circadian mechanism that is independent of transcription. This view was reinforced by the oxidative rhythm of peroxidase later found in non-nucleated red blood cells [14]. In addition, the most famous is the cyanobacteria oscillator based on the phosphorylation principle [19]. There is growing evidence that transcription-based oscillators are not the only means by which cells can track time [60].

Take the cyanobacteria as an example to illustrate the PTO mechanism. Cyanobacteria are the simplest model organisms for studying the biological clock. The mechanism relies on post-translational oscillation consisting of three core clock proteins of KaiA/B/C for rhythmic oscillation [61,62,63]. Among them, the core protein KaiC has autokinase and autophosphatase activity, which can enable KaiC to phosphorylate among its subunits, and it can also automatically dephosphorylate [64]. KaiC exists alone at this time; due to the predominance of autophosphatase activity, it shows spontaneous dephosphorylation. KaiA has a high affinity for KaiC in a low phosphorylation state, which can enhance KaiC autokinase activity, weaken phosphatase activity, and promote KaiC phosphorylation. On the other hand, KaiB binds to KaiC, which is highly phosphorylated and inhibits KaiA, which makes KaiC show a low phosphorylation state [65]. The KaiC protein can maintain rhythmic oscillation under the joint action of KaiA and KaiB proteins alternatively [66].

However, although the KaiA/B/C system of cyanobacteria belongs to the PTO mechanism, it is not a conserved biological clock oscillation system that exists across species. KaiA/B/C homologous genes have been only found in some bacteria [67]. Some studies have used the Mirrortree algorithm to evaluate the co-evolution of the three cyanobacteria KaiA/B/C proteins with the 2-Cys peroxidase family of metabolism/ROS pathways. The results show that the three core components of the cyanobacteria oscillators related to 2-Cys peroxidase have a strong correlation [68,69]. Therefore, some think that the timing mechanism with the PRX protein as the core is considered a more conservative evolutionary clock core marker.

In contrast to the divergent evolution of TTFL, the oxidized PRX presents circadian rhythms in all kingdoms of life, including eukaryotes, cyanobacteria, and archaea [16]. The 2-Cys PRX protein was found to maintain a periodic oxidation-reduction cycle for approximately 24 h without the TTFL mechanism in human mature red blood cells and *Ostreococcus tauri* green algae cells [14,15]. At the same time, this rhythm has the essential characteristics of two biological clocks that can be reset by environmental conditions and temperature compensation [70]. The activity of PRX protease is based on the following processes: 1. Cysteine (PRX-SH) on the PRX protein is oxidized to sulfenic acid (PRX-SOH) under the action of peroxide; 2. PRX-SOH molecule disulfide bonds form dimers (2-Cys PRX); 3. 2-Cys PRX undergoes dimer depolymerization under the action of the sulfur-reducing protein TRX and is reduced to PRX-SH; 4. The cycle restarts. In this catalytic reaction, the oxidized PRX dimer can easily form a polymer macromolecule and cause the cycle to be interrupted. So, there is an “over-oxidation-reduction cycle” in the system: PRX-SOH can be further oxidized by peroxide (PRX-SO_2/3_), which can be reduced by conservative parasulfide redoxin sulfiredoxin (SRX) to supplement PRX-SOH, which can be used as a cell cycle “regulator” [71]. During this process, redox-active cysteine may be peroxidized to Cys-SO_2_H and Cys-SO_3_H, which have been identified as circadian biomarkers. Cys-SO_2_H can be slowly catalyzed into Cys-SOH, and further oxidation to Cys-SO_3_H (known as high oxidation) is considered irreversible [72]. Based on the prediction of the mathematical model, the principle of PRX oscillation is expressed as the combination of a fast PRX peroxidation event and a slow and delayed negative feedback loop, which are the minimum elements required for system oscillation. The relaxation-like oscillations of events like time switch thus highlight the importance of switches in generating oscillations [60].

The specific mechanism of PRX is still unclear, and some studies suggest that different species and even cell tissues have different mechanisms. SRX is involved in the oscillations of mouse adrenal glands, brown adipose tissue, and heart, but no homologues of SRX are present in PRX oscillations in mouse red blood cells, *Caenorhabditis elegans,* and *N. crassa* [73,74]. These findings also reflect the complexity of the PRX system in different species.

Like the TTFL system, the PRX system can regulate physiological functions. Since peroxidation accidentally inactivates PRX, it may be related to H_2_O_2_ signaling [73,75]. It is hypothesized that peroxidized PRX can signal other cellular defense mechanisms, thereby counteracting the production of ROS [76]. Thus, they can directly prevent cell damage [17].

However, the status of PRX’s timing mechanism is still controversial. Some studies suggest that PRX is not a timing mechanism. Due to the time difference between the fast catalytic cycle and the slow peroxidation cycle, the PRX system causes the accumulation of peroxidized and highly oxidized forms (Cys-SO_2/3_H), half of which have a life span of several hours. The rhythmic over-oxidation of PRX can be interpreted as the “memory” of the flux through its catalytic system, so it may not necessarily be its timing mechanism. Consistent with this idea, red blood cells of sulfiredoxin mutant mice still show rhythmic PRX oxidation, and the daily decline in PRX peroxidation appears to be primarily due to proteasome degradation [74]. Currently, PRX is considered to participate in timing in at least three ways: (1) Redox-based oscillations may constitute a truly conservative mechanism for cellular clocks in eukaryotes [77]; (2) Redox and metabolic rhythms are only the post-translational output of the cellular clock function and therefore have nothing to do with the timing mechanism [60]; (3) Changes in metabolism and redox balance possibly constitute a non-essential auxiliary cell clock mechanism. In this case, they can regulate the amplitude and phase of circadian gene expression but do not require oscillations to maintain the timing mechanism of the cell.

Many studies support the first statement, but some studies have found that after disrupting the main metabolic pathways and cell redox balance, cells can still autonomously regulate the circadian rhythm of biological metabolism. It has been found that inhibition of glycolysis alone affects cells, but the clock has no major impact. At the same time, the amplitude and phase of PER protein expression are sensitive to redox balance and oxidative pentose phosphate pathway (PPP) activity, but the oscillation period is insensitive to these perturbations. This suggests that the redox signal is not essential for the cellular timing mechanism, but it serves as the biological clock’s input and output [78]. However, this is a conclusion based on mammalian models, and we have reason to believe that this is due to the performance of the more powerful TTFL mechanism gradually replacing the performance of the PTO mechanism during the evolution of the biological clock.

Notably, the PRX rhythm system was later extended to include mice, insects, worms, bacteria, and even archaea, suggesting that redox circadian rhythms can be a common feature of all aerobic living organisms. This rhythm has nothing to do with the previously identified TTFL clock. The phylogenetic conservatism of the PRX rhythm indicates that the redox oscillator appeared after the major oxidation event 2.5 billion years ago when the ancestor of cyanobacteria advanced the first photosynthetic system and caused a dramatic increase in atmospheric oxygen. The combination of the rhythmic production of oxygen by photosynthetic organisms and the production of ROS by sunlight may promote the common development of clock mechanisms and antioxidant systems [16,79].

### 3.3. Link between the Two Mechanisms

PRX proteins and their oxidative rhythms are highly conserved among species, so PRX rhythms seem evolutionarily ancient. They may have appeared before the circadian TTFL, which is thought to have evolved independently in separate species [16]. Understanding the relationship between TTFL and PRX oscillations is now an important goal, as there is an increasingly obvious interaction between the two (Figure 2). The source of mitochondrial H_2_O_2_ in tissues where redox oscillations have been demonstrated is at least partially under the control of the circadian TTFL. For example, the main source of H_2_O_2_ in the adrenal gland is steroid production, which is regulated by TTFL [80,81]. In the heart or brown adipose tissue with very high oxidative metabolism, the respiratory component becomes an important source of H_2_O_2_, and the breathing rate is rhythmically controlled by TTFL [82]. The cycle of Prx-SO_2_H rhythms in embryonic fibroblasts from *Cry1/2*^-/-^ mice was prolonged [14]. Prolonged PRX oscillations are also shown in fruit flies with mutated rhythms [16].

On the other hand, *Arabidopsis* mutants lacking the *Prx* gene show TTFL rhythms that change their phase or amplitude, and knocking out specific PRX isoforms in human osteosarcoma cells (U2OS) affects the cycle length and amplitude of clock gene rhythms [16,83]. This is supported by the fact that the interaction of two mammalian core clock proteins, PER2 and CRY1, is sensitive to redox [84]. These all indicate that the cell’s redox balance can regulate the TTFL oscillator [85,86].

## 4. Motivation of Biological Clock Evolution

No matter how the biological clock evolved, there is no doubt that it has given biological organisms obvious adaptive advantages. Moreover, these adaptive advantages are divided into the external adaptive advantages of synchronizing external time and the internal adaptive advantages of coordinating internal metabolic processes [87]. The photoperiod and redox cycle are the most important parts of this process. However, as organisms tend to have more complex systems, other factors are involved in the evolution of the biological clock. Because the main PTO system can perform fewer inputs and outputs, performing a larger range of functional control is difficult, so a mechanism that adapts to more complex regulation is bound to appear.

Compared with the PTO system, gene-level transcription and translation are introduced into the TTFL system. The core proteins CLOCK/BMAL1 in the mammalian TTFL system has a wide range of gene promoter properties, which greatly expands the output function of the biological clock. The *Clock*, *Bmal* or *Cycle*, *Per*, and *Cry* or *Tim* genes in the TTFL system can be widely regulated by other molecules, enhancing the input perception of the biological clock. The central biological clock evolved from more complex organisms. In mammals, synchronizing all peripheral clocks via a light-controlled SCN provides many evolutionary advantages. Leguminous leaves have movements driven by changes in circadian rhythm stress. *Arabidopsis thaliana* has other rhythmic movements during hypocotyl and flowering stem elongation [88]. The expression of clock-related genes in different organs is regulated by internal and external signals, increasing the change in expression timing, a trait that may contribute to the plant’s ability to adapt to environmental changes caused by the circadian cycle [55,89,90,91,92]. The expansion of the biological clock function ensures that organisms can better calibrate time [93,94]. More complex circadian clocks enhance input and output functions.

This complex circadian clock regulation has been shown more in advanced life activities. In animals’ cognitive function, a two-way interaction occurs between the circadian system and the memory process, of which more complex learning patterns affect the circadian rhythm entrainment. For a continuous attention task, a rat’s repeated training is not synchronized with the animal’s activity pattern, resulting in a change in the rhythm of circadian motor activity [95,96,97]. Therefore, the biological clock regulates the acquisition and formation of memory, and complex learning tasks and animal activities may also reset the biological clock [98].

In the sexual selection of animals, the choice of core clock components may lead to gender differences, for example, male-specific expressions of specific characteristics, such as antlers, feathers, or courtship performance, or female-specific expression characteristics, such as spouse-choosing behavior, color, or motherhood. This ubiquitous gender difference is more likely to arise from reproductive abilities, such as courtship, territorial establishment, ability to produce offspring, or viability choices. In contrast, sexual selection affects individual differences in one sex, possibly due to clock-controlled genes regulating clock output and gene function [99]. For example, endocrine signals are closely related to reproduction [100]. All this evidence reflects the ability of selection pressure to promote the formation of biological clocks to accept complex inputs and outputs.

## 5. Conclusions

The PTO mechanism is more ancient and primitive, and the TTFL mechanism appeared later. In early life, the PTO mechanism was sufficient to assume the function of a biological clock. However, as life became more and more complex, it was difficult for PTO to manipulate more physiological functions by relying on the mechanism of protein conformation change, so the biological clock began to expand its scale and began to cover more functional pathways. The TTFL mechanism is accompanied by transcription and translation. The ability of the regulator factors to combine with various gene promoters has given the biological clock more powerful input and output functions, and a more powerful biological clock was brought to life. The formation of the TTFL mechanism may have come from evolutionary pressures. Various factors related to time promoted the birth of the new biological clock oscillator so that more advanced biological clocks are mainly based on the TTFL mechanism. Although there is evidence of an evolutionary relationship between the two clocks, more work is needed to determine how the TTFL mechanism evolved. This is of great help for us to clarify the biological clock’s evolution and understand its physiological significance to the biological clock deeply.

## Figures and Tables

**Figure 1 ijms-24-02402-f001:**
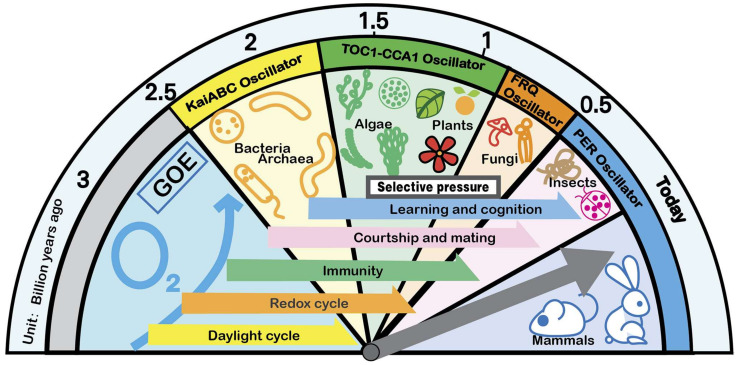
Selective pressure drives the evolution of the biological clock. The biological clocks of various organisms are derived from the selection pressure given by the natural environment. Various selection pressures are the main driving force that drives the biological clock to become more complex and refined. Since the Great Oxygenation Event (GOE) began 3 billion years ago, the biological clock has become more and more complex and manifests in different forms on different species. From the post-translational oscillation (PTO) mechanism to the transcription–translation negative feedback (TTFL) mechanism, the overall biological clock system of life reflects the trend of step evolution.

**Figure 2 ijms-24-02402-f002:**
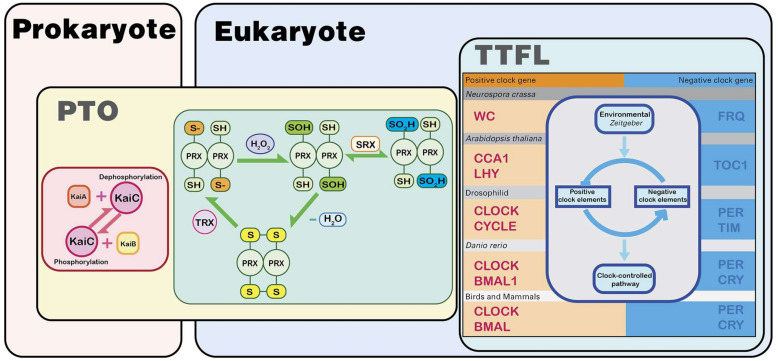
The biological clock in prokaryotes is mainly based on the post-translational oscillation mechanism (PTO). The most typical one is the oscillator with KaiA/B/C as the core, found in cyanobacteria. This oscillator relies on the autokinase and autophosphatase activities of the core protein KaiC for experiments. In eukaryotes, the transcription–translation negative feedback loop (TTFL) mechanism is the main oscillation mechanism of the biological clock. Their molecular composition is different in different organisms, but they have the same principle. There is a biological clock oscillator with the PTO mechanism in eukaryotes. The “redox” oscillator with the PRX protein as the core is widely present in eukaryotes and has a strong correlation with prokaryotic oscillators. The PRX oscillator may be the key to connect the eukaryotic–prokaryotic oscillator and may also be the oldest origin of the biological clock. *Zeitgeber*: an environmental agent (the occurrence of light or dark) that provides the stimulus setting a biological clock.

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
