# Peer review of "Perfecting the Life Clock: The Journey from PTO to TTFL"

_ijms, 2023, doi:10.3390/ijms24032402_

Round 1
Reviewer 1 Report
Two types of regulatory mechanisms are known in biological time keeping and it is crucial to resolve the dualism for evolutionary derivation. The topic is important and interesting to challenge. The manuscript well-summarizes our current understanding on this challenge, though the experimental proof is difficult to obtain. The two figures are helpful but additional one may help the readers to convince on the critical step in transformation from PTO to TTFL for expreimental demonstration in the next round.
Author Response
Thank you very much for reviewing our article! We are also keen to draw the key steps in the transition from PTO to TTFL to complete the manuscript. Unfortunately, we have found very little concrete evidence of a shift between the two. The evolutionary relationship between them is not clear. Few studies have been involved. Many questions remain unanswered. Therefore, we have abandoned drawing this part of the figure to avoid misleading the reader. Thanks again for the review!

Reviewer 2 Report
The manuscript of Weitian Li and colleagues traces a continuous story addressing the changes that circadian system has suffered during the evolution. The biological circadian clocks form a key complex system in all organisms to adapt to the dynamic environment, helping to predict these changes through modifications in physiology and thus stay synchronized. These physiological clocks are well described from prokaryotes to eukaryotes in this review. However, the review is riddled with mistakes and there are several points that the authors need to clarify before publication in International Journal of Molecular Sciences. See all my suggestions in the following detailed report.

Round 2
Reviewer 2 Report
Authors have carried out a great job of revising their work, following perfectly the suggestions given by the reviewers. I do not want to seem harsh, but I have observed few errors on the new document that have to be corrected:
Line 128. Figure 1 caption. It is written "Selection pressure...", while in the figure is written "Selective pressure". I suggest to unify both terms.
Line 147. If I am not wrong, the name of the gene is "Timeless" not "Timless" as authors have written.
Line 373. The word "Zeitgeber" must be written in italics, as it is a German word, like it is into the Figure 2.
Figure 2. The name "Drosophilid" is not a scientific name, so it must be written in normal letter. If authors use "Drosophila" term instead of Drosophilid, it should be written in italics.
